# Conservation of Soil Organic Carbon in the National Park *Santuario de Fauna y Flora Iguaque*, Boyacá-Colombia

**Hernán J. Andrade** [1,2,*], **Milena A. Segura** [1,3] **and Diana S. Canal-Daza** [1,2]

1    Research Group of Eco-Friendly Production of Tropical Crops—"PROECUT",
     Ibagué Tolima 730006299, Colombia
2    Department of Production and Plant Health, Agronomic Engineering Faculty, Universidad del Tolima,
     Ibagué Tolima 730006299, Colombia
3    Department of Forest Sciences, Forest Engineering Faculty, Universidad del Tolima,
     Ibagué Tolima 730006299, Colombia
*    Correspondence: hjandrade@ut.edu.co

**Abstract:** Protected areas are important zones for the conservation of strategic ecosystems that provide environmental services to human populations. The *Santuario de Fauna y Flora Iguaque* (SFFI) (Boyacá, Colombia) preserves an important area of páramos and andean high-land forests that offer water and other services. Soil organic carbon (SOC) was estimated at a depth of 0–30 cm in the four dominant land uses: (1) natural grasslands prevailingly without trees and shrubs (NSWT), (2) broad-leaved forests with continuous canopy, not on mire (BFCC), (3) open heathlands and moorlands (OMH), and (4) dense heathlands and moorlands (DMH). This classification is based on Corine Land Cover, adapted for Colombia. Land uses did not differ significantly ($p > 0.05$) in the SOC stock, with values of 139.0, 131.1; 101.1; and 83.0 Mg C/ha in OMH, BFCC, NSWT, and DMH, respectively. In total, SFFI retains 593 Gg C in that soil layer. Projections of effects caused by potential land use changes show that up to 461.0 Gg $CO_2$ could be transferred to the atmosphere if this conservation area is not preserved. SFFI, due to its conservation strategies, allows storing significant amounts of atmospheric carbon and becomes an effective strategy of climate change mitigation.

**Keywords:** bulk density; climate change; land use and land use change; mitigation





## 1. Introduction

Climate change is a worldwide problem due to the large amounts of greenhouse gases (GHG) emitted to the atmosphere as a product of human activities [1,2]. Since the pre-industrial era, atmospheric concentration of GHG has increased significantly [3]. Currently, the concentration of carbon dioxide ($CO_2$) is around 417.1 ppm [4], mostly due to the economic and demographic growth of human populations [1]. Atmospheric $CO_2$ reached 147% of what it was in the pre-industrial era in 2018 [5]. Latin America and the Caribbean are responsible for 8.3% of the total $CO_2$ worldwide, excluding land use change; with estimations of a total emissions worldwide of 49 Pg $CO_2$e/year [6,7]. IPCC [8] states that, 13% of global $CO_2$ emissions between 2007 and 2016 is caused by agriculture, forestry, and activities related with other land uses (AFOLU). At the same time, AFOLU represents 23% of the net total anthropogenic emissions of GHG, this is the reason why it is the main focus of mitigation and reduction projects.

Soils contain more carbon (C) than the joint stock in vegetation and the atmosphere, and it can be found organic and inorganically [9–11]. Soil organic carbon (SOC) has been estimated through several methods, and its value is around 1500 Pg at 1 m depth, thus being an important component of the C global cycle (69.8% of the organic C in the biosphere) [12–15]. On the other hand, historic global C losses due to agricultural activities are estimated to be around 55–100 Pg from the SOC reserve, and 100–150 Pg from the biotic carbon pool [14,16].

Mitigation is led through actions that allow the reduction of emission sources, boost GHG sinks, and therefore, reduce climate change on the long term [17–19]. Carbon sequestration and storage in soils and biomass is presented as another option on the list of actions to reduce or stabilize the increase in GHG atmospheric concentrations [20,21]. Other options involve improving energetic yield, switching to less carbon-intensive fuels, and using renewable energy sources [22]. According to FAO [13], improving soil management is one of their relevant points, as it would guarantee food safety for the current world population and, additionally, it would store C in soils to minimize climate change threats.

Páramos make a very special life zone worldwide, and they are very important for Andean countries due to their biological, hydrological, social, economic, and cultural interest [21,23–25]. In Colombia, the páramos covers around 14,000 km$^2$, which makes for 1.3% of the national territory, with the departments of Boyacá (18.3%), Cundinamarca (13.3%), Santander (9.4%), Cauca (8.1%), Tolima (7.9%), and Nariño (7.5%) as the ones with the most cover [26]. Colombian National Park System (NPS) consists of 49 protected areas and covers almost 10 million hectares (9% of the national territory), and its goal is to guarantee the preservation of natural resources, biodiversity conservation and stability of ecological processes for human development [27]. The *Santuario de Fauna y Flora Iguaque* (SFFI) is part of the NPS.

The objective of this study is to estimated SOC stock in the dominant land uses of the SFFI in Boyacá (Colombia), the results would be key tools to understand the importance of this protected area in terms of C sinks and the potential changes in SOC reserves due to land use changes. The research hypotheses were: (1) there are significant differences in BD and SOC concentration and stock between land uses; (2) the increase in SOC carbon concentration presents benefits to the soil, using BD as an indicator; (3) some land use changes would cause large $CO_2$ emissions to the atmosphere.

The results can be used to promote policy development for the conservation of these areas. The novelty of this research is the estimation of the total SOC stock and the changes, such as carbon increment or $CO_2$ emission, that could occur if land use is changed.

## 2. Materials and Methods

### 2.1. Area of Study

The *Santuario de Fauna y Flora Iguaque* (SFFI) is located in the central-western area of the East Mountain Range in Colombia, between the Departments of Boyacá and Santander, in the municipalities of Arcabuco, Chíquiza, and Villa de Leyva [28,29]. This area is known as the corridor of páramos and forests of Iguaque-Guantiva-La Rusia, where SFFI is located in the southern end [30]. This protected area is located between the coordinates 5°44′29.90″ to 5°35′52.69″ N, and 73°31′8.18″ to 73°22′43.38″ W, it has diverse elevations from 2400 to 3800 m, a mean annual precipitation of 1648 mm, average temperature of 13 °C, and different land uses along 6923 ha (Figure 1).

### 2.2. Sampling Design

An unbalanced stratified sampling design was used, evaluating four land uses: (1) natural grasslands prevailingly without trees and shrubs (NSWT); (2) broad-leaved forests with continuous canopy, not on mire (BFCC); (3) open heathlands and moorlands (OMH); and (4) dense heathlands and moorlands (DMH) (Table 1). This label was given according to the classification of Corine Land Cover, adapted for Colombia [31]. The number of repetitions in each land use depended on their total area in the SFFI, in a way that increases representability (Table 1). After that, total SOC stock was estimated for the whole protected area, considering the area of each land use. The soils of the SFFI are divided in two large groups: piedmont soils and hillside soils. The hillside soils are coarse in structure (sandy loam) to very coarse (loamy sandy), that belong to the Lithic Udorthents and Typic Dystrudepts. In contrast, piedmont soils are well-drained from medium (clay loam) to fine (clay) structures [29].

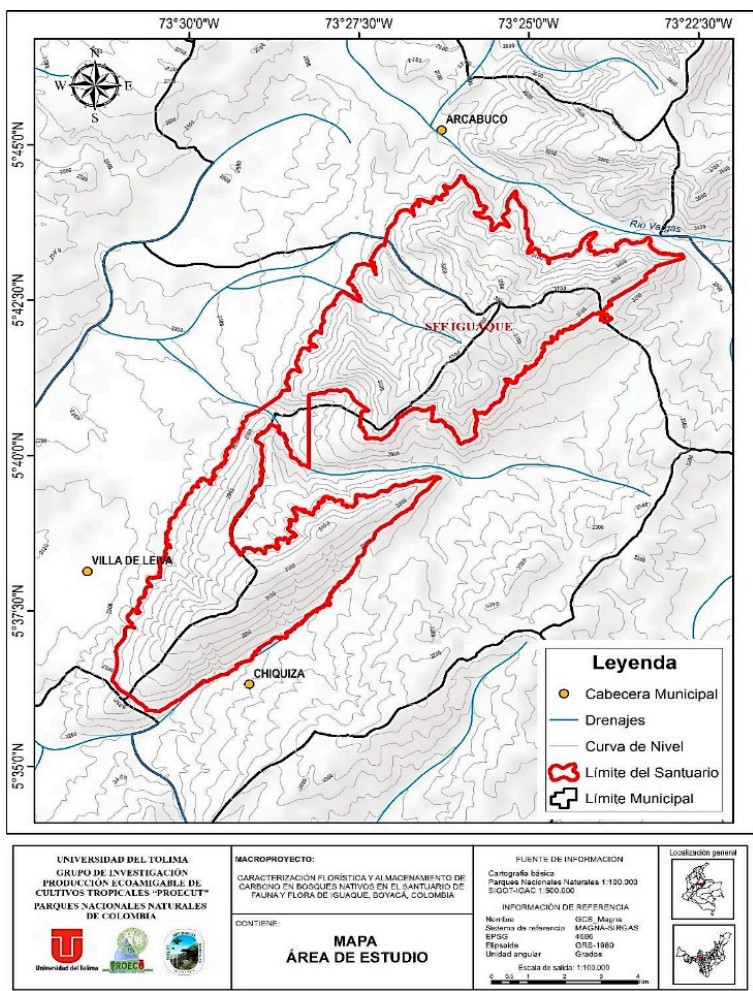

**Figure 1.** Location of the study area, *Santuario de Flora y Fauna Iguaque* (SFFI), Boyacá, Colombia. Source: PROECUT Research Group.

**Table 1.** Land uses in *Santuario de Fauna y Flora Iguaque*, Boyacá, Colombia.

| Land Use | Area (Ha) | Repetition Number |
|---|---|---|
| Natural grassland prevailingly without trees and shrubs (NSWT) | 1755.8 | 17 |
| Broad-leaved forest with continuous canopy, not on mire (BFCC) | 1477.8 | 18 |
| Open heathlands and moorlands (OMH) | 1227.8 | 9 |
| Dense heathlands and moorlands (DMH) | 595.1 | 5 |
| Total | 5056.5 | 49 |

### 2.3. Estimation of Soil Organic Carbon (SOC) Stock

SOC stock was determined through soil samples collected at a 0–30 cm depth, following the IPCC recommendations [32] and Andrade et al. [33]. Bulk density (BD) was determined using a graduated cylinder (98.2 cm$^3$), collecting three soil samples per repetition [32,33], which were dried in an oven at 105 °C until constant weight was reached. Bulk density is determined as the ratio between the dry weight of the soil and the internal volume of the cylinder. SOC concentration was determined from a composed sample constituted by 10 sub-samples each, using the spiral auger. After that, soil samples were analyzed at Láserex Laboratory from Universidad del Tolima (Tolima, Colombia), using

the Walkley & Black method [34]. The proportion of gross fragments was not considered, given that it was observed during sampling that this variable did not exceed 5% in volume.

### 2.4. Impact of Land Use Changes in SOC

Projections of the potential variations in SOC stock attributed to land use changes were made. In this case, SOC stock estimations were carried out considering soil mass and not soil volume to avoid the effects of changes in BD that would alter the SOC results, as suggested by different studies [10,35–37]. For this, the lowest BD found was used and considered as the baseline, where the impacts on SOC stock was due to changes in SOC concentration. SOC stock values were multiplied by 3.67 to estimate $CO_2$. These projections were made in terms of carbon density per area unit and in total for SFFI. In the case of unit values, the difference in SOC stock between future and current land use was calculated. Total projections were calculated for the area of each land use; calculating what would happen to the SOC stock if the entire area of the current land use were to change to another land use. This was calculated as the difference between the SOC stock in the future use and the current use, using the total area of the current use. These results generate additional carbon sequestration values in case of positive values, or possible $CO_2$ emissions in case of stock reduction [33,36,38]. It was estimated the $CO_2$ emissions if the three land uses with the highest SOC stock change to that with the lowest carbon stock.

### 2.5. Data Analysis

Normality of variables studied was tested using modified Shapiro–Wilks test. The same way, an analysis of variance (ANAVA) was made to those variables that showed normal distribution, while those that were not normal, such as SOC concentration, underwent a logarithmic transformation. Land use means were compared with Fisher test to observe if there were statistical differences among variables tested in case of being detected by the ANAVA. Statistical analyses were carried out using the InfoStat software with a 95% confidence.

## 3. Results

### 3.1. Bulk Density (DB) and Soil Organic Carbon (SOC) Concentration

BD showed significant differences ($p < 0.05$) among the different land uses (Figure 2). BFCC showed a significantly ($p < 0.05$) higher BD (0.88 Mg/m$^3$) than the other land uses tested (0.72, 0.66 and 0.52 Mg/m$^3$ for OMH, DMH, and NSWT, respectively). Areas without trees and low abundance of shrubs (NSWT and OMH) showed a significantly higher ($p < 0.05$) value of SOC (6.5–6.7%) that those areas with trees or high abundance of shrubs (BFCC and DMH) with values between 4.2 an 4.8% (Figure 2).

### 3.2. Relationship between BD and SOC Concentration

An inverse relationship between BD and SOC concentration in the land uses evaluated was found (Figure 3). This means that, as SOC concentration increases, BD decreases, thus resulting in a reduction in soil compaction and the improvement of other properties such as aeration and retention and movement of water.

### 3.3. SOC Stock

No significant differences ($p > 0.05$) were detected in SOC stock among land uses in the top 30 cm of soil. However, the highest value was found in OMH (139.0 Mg/ha), followed by BFCC, NSWT, and DMH, with 131.1, 101,1 and 83.0 Mg/ha, respectively (Figure 4). SFFI had a total of 591.3 Gg of SOC in the top 30 cm of soil. These results show the importance of forest conservation and other natural land uses in protected areas, such as SFFI, as they store large amounts of SOC.

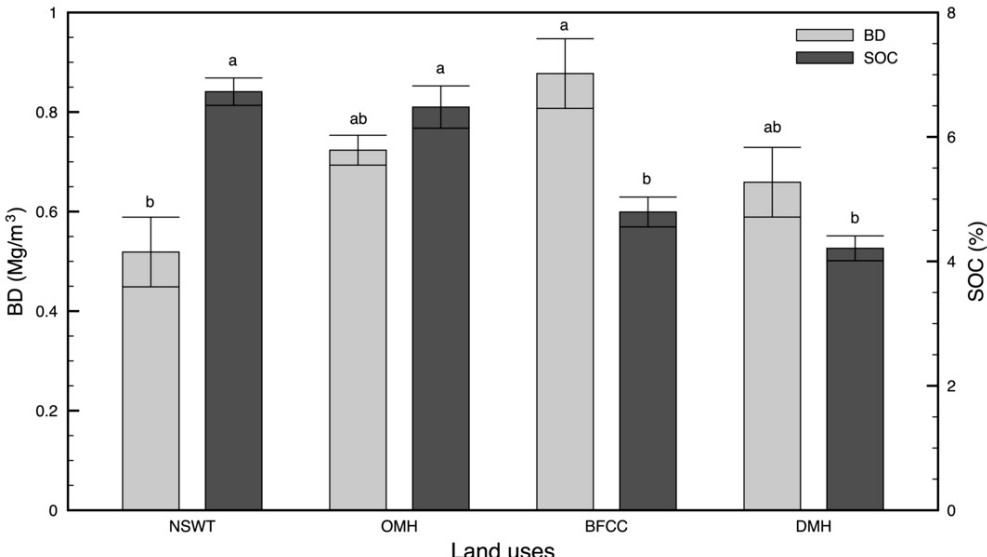

**Figure 2.** Bulk density (BD) and soil organic carbon (SOC) concentration at a 0–30 cm depth, in the four dominant land uses at *Santuario de Fauna y Flora Iguaque*, Boyacá, Colombia. NSWT: natural grasslands prevailingly without trees and shrubs; OMH: open heathlands and moorlands; BFCC: broad-leaved forests with continuous canopy, not on mire; DMH: dense heathlands and moorlands. Error bars correspond to standard error. Different letters indicate statistical differences ($p < 0.05$).

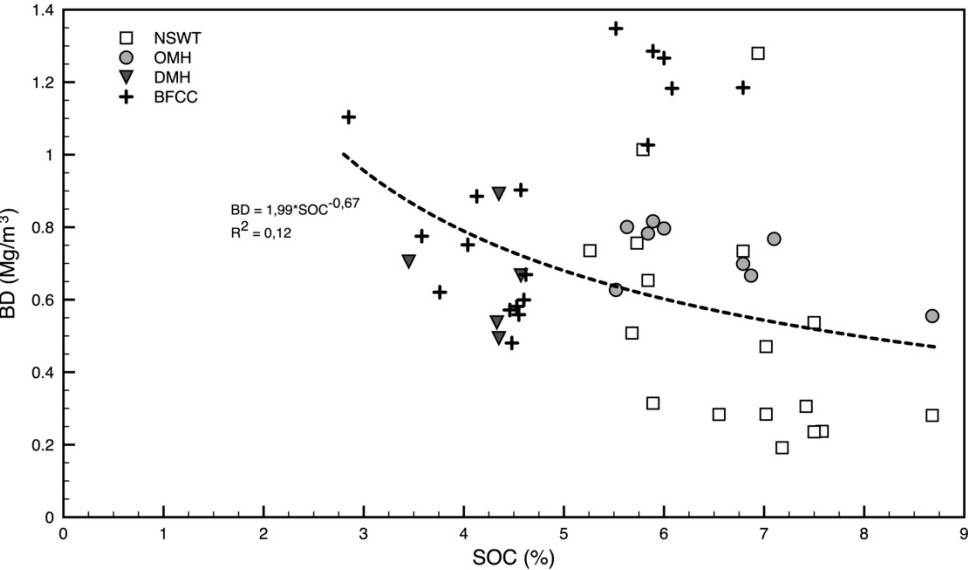

**Figure 3.** Relationship between soil organic carbon (SOC) concentration and bulk density (BD) at a 0–30 cm depth of the soil, in the four dominant land uses at *Santuario de Fauna y Flora Iguaque* in Boyacá, Colombia. NSWT: natural grasslands prevailingly without trees and shrubs; OMH: open heathlands and moorlands; BFCC: broad-leaved forests with continuous canopy, not on mire; DMH: dense heathlands and moorlands. Pointed line represents the best fit model.

*3.4. Impact of Land Use Changes in SOC Stock*

Considering just SOC, the most favorable land use change would be to exchange DMH for NSWT or OMH, which would allow a carbon additionality of 143.8 and 127.7 Mg $CO_2$/ha, respectively (Figure 5). Meanwhile, inverse changes would cause the highest $CO_2$ emissions. The most notorious change would take place when changing the 595.1 ha from DMH to OMH, which would cause a total carbon additionality of 504.9 Gg $CO_2$. In contrast, the most harmful change, from the perspective of climate change mitigation, would be that the whole NSWT area (1755.8 ha) were changed into DMH, emitting around 254.3 Gg $CO_2$ to

the atmosphere (Figure 5), and contributing to rise climate change. If the three land uses with the highest SOC stock (OMH, BFCC and NSWT) change to that with the lowest SOC, a total of 461.0 Gg $CO_2$ would be emitted to the atmosphere.

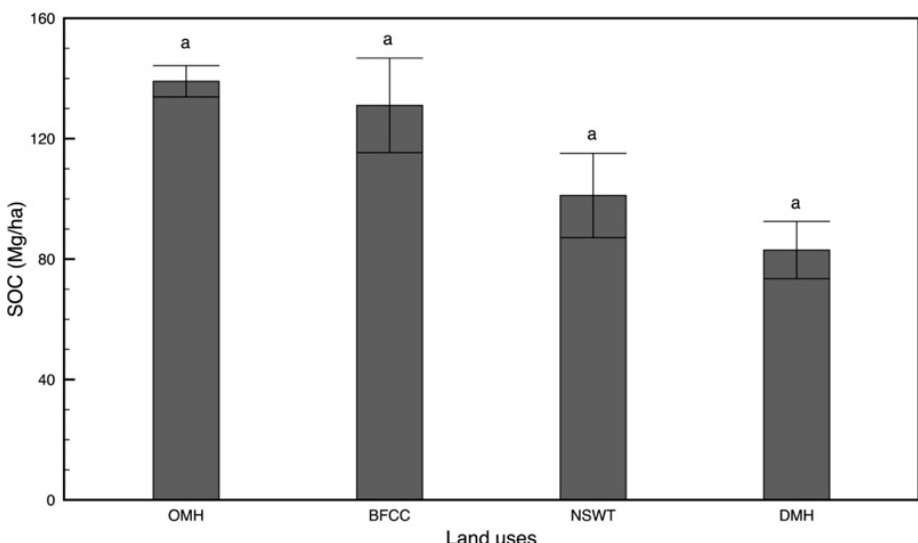

**Figure 4.** Soil organic carbon (SOC) stock at a 0–30 cm depth in the four dominant land uses at *Santuario de Fauna y Flora Iguaque*, Boyacá, Colombia. OMH: open heathlands and moorlands; BFCC: broad-leaved forests with continuous canopy, not on mire; NSWT: natural grasslands prevailingly without trees and shrubs; DMH: dense heathlands and moorlands. Error bars correspond to standard error. Equal letters indicate that there are no statistical differences ($p > 0.05$) among land uses.

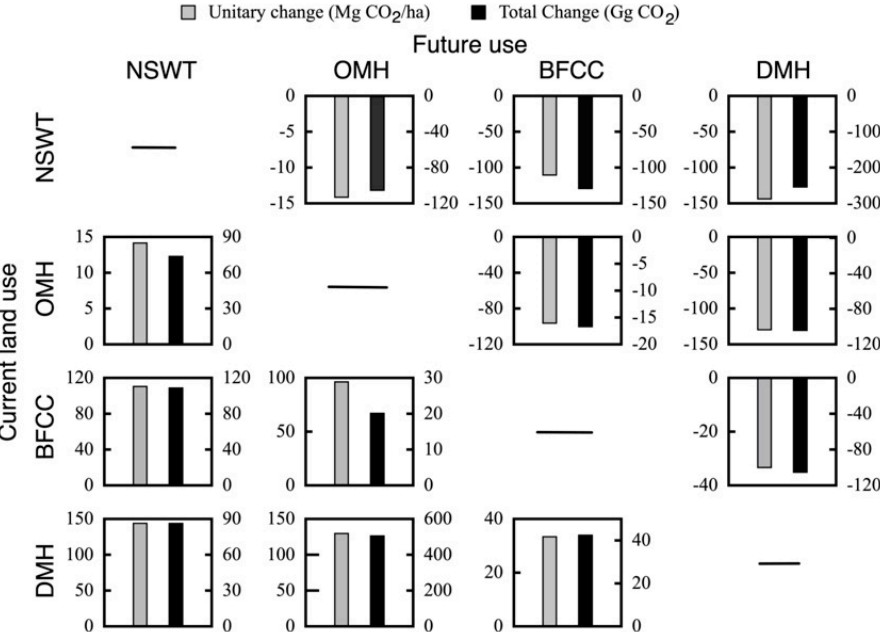

**Figure 5.** Unitary and total change in soil organic carbon (SOC) stock at a 0–30 cm depth due to possible soil use changes at *Santuario de Fauna y Flora Iguaque*, Boyacá, Colombia. NSWT: natural grasslands prevailingly without trees and shrubs; OMH: open heathlands and moorlands; BFCC: broad-leaved forests with continuous canopy, not on mire; DMH: dense heathlands and moorlands. Positive values indicate an increase in SOC stock, whereas the negative show a reduction with the consequent $CO_2$ emission.

## 4. Discussion

### 4.1. Bulk Density (BD) and Soil Organic Carbon (SOC) Concentration

BD values indicate the little effect of these land uses and a high quality of soils by having adequate drainage, porosity and aeration [10,39,40]. These results match the findings of Díaz et al. [41], who estimated a BD of 0.88 Mg/m$^3$ at the same depth for primary forests in San Martín—Perú. Just the same, Rojas et al. [36] found similar values in native forests of Santa Isabel (Colombia) with 0.65 Mg/m$^3$. In Boyacá, BD values of 0.43 and 0.54 Mg/m$^3$ were reported at 0–15 cm and 15–30 cm depth, respectively [42], values that match those found on this current study. The BD registered for NSWT shows values around the ones estimated by Andrade et al. [43] in this same type of land use in the region of Anaime, Tolima (Colombia): 0.46 Mg/m$^3$. Viana et al. [44] found the highest BD in degraded areas, whose values increase with restoration strategies and in native forests from the Amazon.

SOC concentration found in this study is below the mean value reported by Gutiérrez et al. [45], of 11.3% for páramo ecosystems in Colombia. In contrast, Andrade et al. [43] reported a 2.9% SOC in high-land andean forests in the Páramo of Anaime, Tolima (Colombia). The BD in páramos can be highly variable and depends mainly on the humidity conditions of the region; in the dry páramos, this value is between 0.6 and 0.9 Mg/m$^3$ [46], values that agree with the results of this investigation. The small differences in the BD among land uses show the low anthropic impacts in this conservation area.

### 4.2. Relationship between BD and SOC Concentration

The inverse relationship found between BD and SOC concentration, which is win–win situation, has been detected in other studies [36,47,48]. This is attributed to floristic diversity and organisms present in native land uses, which favor the increase of soil organic matter and the SOC [10]. Organic matter from senescence of plant biomass, such as fine roots, leaves and branches, increases soil volume and porous space, causing a substantial improvement in soil quality [43,49,50]. Viana et al. [44] found correlation values between BD and soil organic matter (in direct relation with SOC) very similar to the findings of this current study (r = −0.85). These results show the double positive impact of SOC sequestration by improving not only organic matter, but also increasing porosity, aeration and water retention and movements [51–53] by reducing BD.

### 4.3. Soil Organic Carbon (SOC) Stock

Andean páramos can retain between 119 and 397 Mg/ha SOC in the top 40 cm of soil, a value that can be increased in the case of marshes [54]. Findings on this study are greater than others in Colombia, such as Fernández et al. [42], who estimated in 106 Mg/ha the SOC in Páramo of Rabanal in Boyacá; or Ordoñez et al. [55] in Andean soils in Cauca (111 Mg/ha), but rather below the 188 Mg/ha calculated for Páramo de Sumapaz in Cundinamarca [56]. In contrast, Andrade et al. [43] reported that native forests from Páramo of Anaime in Tolima (Colombia) store just 21.6 Mg/ha SOC at the same depth.

Dieleman et al. [57] argued that SOC stock increases with altitude. However, Phillips et al. [58] show that there are no differences in SOC of Colombian Andean forests when different altitude ranges are compared, reporting that these ecosystems can store up to 124.7 Mg C/ha in the top 100 cm of the soil. The SFFI stores in total 591.3 Gg SOC in the top 30 cm of the soil, and part of this carbon could get lost and return into the atmosphere as $CO_2$, if the purpose of the land use were changed from conservation of the protected area and turned into an agricultural area [59]. Shukla et al. [60] states that SOC is the most important indicator of soil quality, showing that high levels of this variable indicate high-quality soils.

### 4.4. Impact of Land Use Changes in SOC Stock

The impact of land use changes on SOC stock shows the importance of conservation and planning strategies for these protected areas, looking forward to improving and

optimizing the ecosystem services they provide [61–63]. SOC is not the only carbon component that these ecosystems can store. Biomass and litter can complement these contributions [38]. Complementary research to this study showed that SFFI stores 135.9 Gg C in its biomass [64]. Therefore conservation, restoration, management and improvement of soils can increase the accumulation of SOC, playing an important role in mitigating climate change [59]. This carbon conservation capacity in protected areas makes them important areas to be included in Reducing Emissions from Deforestation and Forest Degradation (REDD+) projects of developing countries to offer incentives to contribute with climate change mitigation [65–67]. Other actions within the frame of REDD+ projects could offer other several ecological, social and financial benefits [68,69].

The main limitation of the study is that the timing of SOC stock change according to potential land use changes is not known. The main implication of the results of this study is the knowledge of the carbon stock and the impacts of land use changes, so that conservation can be prioritized by planners and managers.

## 5. Conclusions

Land uses studied at SFFI are very important as atmospheric carbon sinks given that they all have the capacity of storing significant amounts of SOC (83.0 to 139.0 Mg C/ha). This is an additional reason to promote the creation and preservation of protected areas that allow conservation of ecosystems and the environmental services they provide. Apart from the climate change mitigation due to SOC sequestration, the incorporation of atmospheric carbon to soils causes a BD reduction, which brings a better soil quality as a result.

The SFFI stores 591.3 Gg COS in the top 30 cm of soil, which justifies its status as a protected area, since it would promote carbon sequestration and reduce emissions due to deforestation and degradation as one of the ecosystem services that it can provide. Conservation strategies in other areas and land uses must also be implemented in such a way that carbon stored in their components can be conserved, given that some land use changes can cause the return of significant amounts of $CO_2$ into the atmosphere. Likewise, despite the change from OMH to NSWT would cause a positive change in SOC, as the amount of carbon stored would increase and carbon credits could be claimed, other carbon components and other ecosystem services offered must be analyzed.

**Author Contributions:** Conceptualization, H.J.A. and M.A.S.; methodology, H.J.A., M.A.S. and D.S.C.-D.; writing—original draft preparation, D.S.C.-D.; writing—review and editing, H.J.A. and M.A.S.; project administration, M.A.S.; funding acquisition, H.J.A. All authors have read and agreed to the published version of the manuscript.

**Funding:** This study was funded by the Central Research Committee from Universidad del Tolima, through project code 530114.

**Institutional Review Board Statement:** Not applicable.

**Informed Consent Statement:** Not applicable.

**Acknowledgments:** The authors thank the Central Research Committee from Universidad del Tolima. They also thank the *Santuario de Fauna y Flora Iguaque* National Park.

**Conflicts of Interest:** There is no conflict of interest that represents a risk to the validity of the results presented.

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
