# Peer review of "Conservation of Soil Organic Carbon in the National Park Santuario de Fauna y Flora Iguaque, Boyacá-Colombia"

_forests, doi:10.3390/f13081275_

Round 1

Reviewer 1 Report

- It is very important to describe in materials and methods the representive soil types in the study area. This helps to discuss the results.

- Explain how the determnation (not estimation) of bulk density was done using a graduated cylinder

- Soil organic carbon concentration was determined, not estimated

- Explain how soil organic stock was determined considering soil mass instead soil volume

- It is important explain (in materials and methods) which land use change projections were made, in order to better explain the results presented in figure 5.

- There are many observation in references

Author Response

-It is very important to describe in materials and methods the representive soil types in the study area. This helps to discuss the results.

R/ Description of soils in SFFI was included.

-Explain how the etermination (not estimation) of bulk density was done using a graduated cylinder

R/ Explanation of the approach was completed.

-Soil organic carbon concentration was determined, not Estimated

R/ We agree with the suggestions. Done.

-Explain how soil organic stock was determined considering soil mass instead soil volume

R/ A better explanation was included in the section. A new reference was included.

-It is important explain (in materials and methods) which land use change projections were made, in order to better explain the results presented in figure 5.

R/ An explanation of the projections was added. The most important results of Figure 5 are described immediately before, we think more details can be seen on that.

-There are many observation in references

R/ Thank you. Corrections were incorporated.

-Corrections at pdf file

R/ All corrections/suggestions done in the pdf file were considered; most of them were done.

Reviewer 2 Report

Dear authors,

In general, I think the paper has interesting results and could be published. A few suggested changes are listed, but other than these small typos there is a lot to recommend publishing this paper.

General comments:

- The quality of English needs to be improved in some parts (mainly abstract, introduction and methods).

- The number and quality of references in the Introduction and discussion are poor. I am sure there are more relevant studies to your work (see my comments and example citations in specific comments). Also, it must be discussed why protected areas are important? 

- The novelty of the paper must be better emphasized.

- The arrangement of the structure of the manuscript is inappropriate. For example, in the introduction, there is no coherency among paragraphs. This arrangement is confusing and would greatly weaken the reasonableness of this study.

- At the end of the discussion, add a research limitation and implication of your study for planners and managers.

- Specific comments

- L 25: Revise "land use and land use change" to "land use and land cover change".

- L 54-55: There is no connection between these paragraphs. 

- L 65: Research hypothesis and research questions should be added.

- In discussion (and also introduction), these papers can be considered by authors:

** https://doi.org/10.1016/j.agee.2021.107326

** https://doi.org/10.3390/f12121805

** https://doi.org/10.1111/gcb.15998

** https://doi.org/10.1016/j.scitotenv.2019.02.420

** https://doi.org/10.1016/j.cosust.2020.12.005

** https://doi.org/10.1016/j.catena.2021.105227

** https://doi.org/10.1016/j.scitotenv.2018.01.104

** https://doi.org/10.1007/s10342-018-1138-8

Author Response

Specific comments

-L 25: Revise "land use and land use change" to "land use and land cover change"

R/ Throughout the document we are working with land use and land use change. We could change all the uses to covers, but we prefer to leave it this way to be congruent.

L 54-55: There is no connection between these paragraphs.

R/ It was done. Latter paragraph was modified to improve connection.

L 65: Research hypothesis and research questions should be added.

R/ Research hypothesis were added.

In discussion (and also introduction), these papers can be considered by authors:

R/ Four references were added.

General comments

- The quality of English needs to be improved in some parts (mainly abstract, introduction and methods).

R/ This was checked, considering comments from reviewer 1. We would appreciate it if you could point out some grammatical errors.

- The number and quality of references in the Introduction and discussion are poor. I am sure there are more relevant studies to your work (see my comments and example citations in specific comments). Also, it must be discussed why protected areas are important?

R/ Four references recommended by the reviewer 2 were incorporated. Discussion includes the importance of protected areas in terms of carbon and other ecosystem services.

- The novelty of the paper must be better emphasized.

R/ It was done in introduction.

- The arrangement of the structure of the manuscript is inappropriate. For example, in the introduction, there is no coherency among paragraphs. This arrangement is confusing and would greatly weaken the reasonableness of this study.

R/ Coherence of the manuscript was checked again, as reviewer recommended. We believe introduction structure is coherent: GHG emissions, Latin America an AFOLU contribution, importance of soil in global carbon, mitigation, importance of páramos and other high-land Andean ecosystems and objective, relevance and novelty of the study.

- At the end of the discussion, add a research limitation and implication of your study for planners and managers.

R/ We agree. This was added to discussion.

Round 2

Reviewer 2 Report

Dear authors,

Good job!